# Isotope engineering achieved by local coordination design in Ti-Pd co-doped ZrCo-based alloys

Jiacheng Qi[1,5], Xu Huang[2,5], Xuezhang Xiao ®[1,3] ✉, Xinyi Zhang[1], Panpan Zhou[1], Shuoqing Zhang[1], Ruhong Li ®[4], Huaqin Kou ®[2] ✉, Fei Jiang[2], Yong Yao[2], Jiangfeng Song[2], Xingwen Feng[2], Yan Shi[2], Wenhua Luo[2] & Lixin Chen ®[1] ✉

Deuterium/Tritium (D/T) handling in defined proportions are pivotal to maintain steady-state operation for fusion reactors. However, the hydrogen isotope effect in metal-hydrogen systems always disturbs precise D/T ratio control. Here, we reveal the dominance of kinetic isotope effect during desorption. To reconcile the thermodynamic stability and isotope effect, we demonstrate a quantitative indicator of $T_{gap}$ and further a local coordination design strategy that comprises thermodynamic destabilization with vibration enhancement of interstitial isotopes for isotope engineering. Based on theoretical screening analysis, an optimized Ti-Pd co-doped $Zr_{0.8}Ti_{0.2}Co_{0.8}Pd_{0.2}$ alloy is designed and prepared. Compared to ZrCo alloy, the optimal alloy enables consistent isotope delivery together with a three-fold lower $T_{gap}$, a five-fold lower energy barrier difference, a one-third lower isotopic composition deviation during desorption and an over two-fold higher cycling capacity. This work provides insights into the interaction between alloy and hydrogen isotopes, thus opening up feasible approaches to support high-performance fusion reactors.

The deuterium-tritium (D-T) fusion reaction that can produce substantial net energy is currently one of the most promising ways to tackle the energy crisis[1-4]. However, the development of fusion energy suffers from low fuel burn efficiency, due to the decoupled distributions and distinct transport behaviors of D/T particles[5-9]. The key to achieve self-sustaining and long-term D-T fusion reaction, thus, lies in fueling with precise D/T ratio. To obtain high-purity and required DT ratio from exhausted fluxes to fuel the plasma, the inner fuel cycle (IFC) of a fusion reactor includes a tokamak exhaust processing system (TEP), an isotope separation system (ISS) and a storage and delivery system (SDS). Owing to the design requirements of a fusion reactor for rapid fueling and accurate isotopic composition, major exhaust gases

are processed through the TEP and SDS while the remaining gases are allowed to transfer to the ISS for separation of deuterium and tritium. To this end, the SDS equipped with metal hydride getter beds is designed to achieve the rapid storage of unreacted DT gas and the gas delivery with controllable composition[10,11]. Nevertheless, the precise control of fuel D/T ratio is always disturbed by the prominent hydrogen isotope effect that originates from the isotopic affinity difference between gas and solid phases[12]. A list of abbreviations and their corresponding detailed description mentioned in this paper are presented in Supplementary Table 1 to improve the readability.

Separation factor ($\alpha$)[13,14], defined by the ratio of equilibrium isotopic composition in the gas and solid phases (Eq. (1)), is widely used to

¹State Key Laboratory of Silicon and Advanced Semiconductor Materials, School of Materials Science and Engineering, Zhejiang University, Hangzhou 310058 Zhejiang, China. ²Institute of Materials, China Academy of Engineering Physics, Mianyang 621907 Sichuan, China. ³Key Laboratory of Hydrogen Storage and Transportation Technology of Zhejiang Province, Hangzhou 310027 Zhejiang, China. ⁴ZJU-Hangzhou Global Scientific and Technological Innovation Center, Zhejiang University, Hangzhou 311215, China. ⁵These authors contributed equally: Jiacheng Qi, Xu Huang. ✉e-mail: xzxiao@zju.edu.cn; kouhuaqin@caep.cn; lxchen@zju.edu.cn

describe the interphase isotopic distribution difference.

$$\alpha = \frac{(Q_l/Q_h)_{gas}}{(Q_l/Q_h)_{solid}} \qquad (1)$$

where $(Q_l/Q_h)_{gas}$ and $(Q_l/Q_h)_{solid}$ are the atomic fraction of light and heavy isotopes in the gas and solid phases, respectively. Early work shows that the separation factor involving mixed isotopes at certain temperature can be calculated by the ratio of the square root of plateau pressure, determined by Van't Hoff relationship for single-isotope isotherms (Detailed discussion in Supplementary Note 1). Thus, accurate determination of thermodynamic parameters is crucial for understanding the temperature dependence of isotope effect.

To meet the requirements of SDS, the hydrogen isotope storage candidate prefers ZrCo alloy than depleted uranium (DU) due to the merits of high hydride stability (-$10^{-3}$ Pa at 20 °C), non-radioactive character, low pyrophoric nature and strong ability of trapping $^3$He[15–21]. Early efforts in ZrCo-based alloys focused mainly on improving the cycling stability through inhibiting the hydrogen-induced disproportionation reaction (HID)[15,21–23]. In recent years, research has expanded to encompass the hydrogen isotope effect of ZrCo-based alloys. Previous researches indicate that the thermodynamic isotope effect of ZrCo alloy exhibits obvious temperature dependence, the plateau pressure for light isotope outweighing that for heavy isotope at a relatively low temperature while the situation could be inverted at an elevated temperature, corresponding to negative and positive isotope effects, respectively. Yet, the reported thermodynamic parameters were scattered (Supplementary Fig. 1)[15,16,24–26], which would exponentially influence the separation factor and render the accurate thermodynamic isotope effect unattainable.

Moreover, the kinetic isotope effect during rapid storage and delivery processes can be evaluated by thermal analyses and real-time monitoring of gaseous isotopic composition[27–32]. The noticeable fluctuation of isotopic composition during desorption but negligible during absorption with various H/D ratio were observed for ZrCo-based alloys at practical scales, which hinders the stable supply with desirable isotopic ratio[27–30,33]. Owing to the prominent kinetic isotope effect, the isotopic composition deviation could be up to 7.30% during temperature programmed desorption with initial H/D = 1/1 in a full-scale ZrCo bed assembly[33].

Bridging the gap between the operating temperature ($T_{1\ bar}$) for atmospheric pressure delivery and the critical temperature ($T_{cr}$) for zero isotope effect is expected to make impressive gains in delivery performance. It is highly desirable to reduce $T_{1\ bar}$ for narrowing the temperature gap and concurrently improving anti-disproportionation ability and cycling stability. On the other hand, the atomic mass distinction among hydrogen isotope atoms results in the zero-point energy (ZPE) difference in both gas and solid phases, dictating the interphase isotopic distribution and further $T_{cr}$[34,35]. More importantly, $T_{1\ bar}$ and $T_{cr}$ show strong sensitivity with the local environment of interstitial isotopes, directly linked to the interstitial coordination atoms. Element substitution, therefore, stands out as a facile strategy for tailoring interstitial coordination environment among modification methods including nanostructuring, surface treatment and mechanical ball-milling[36–54]. Although a myriad of efforts have been devoted to reducing the hydrogen isotope effect, the trial-and-error strategy still dominates the alloy development. An effective guiding principle for the alloy design is indispensable.

In this work, the thermodynamic and kinetic isotope effects of ZrCo alloy are precisely obtained. A significant kinetic isotope effect during desorption whereas a negligible one during absorption is observed. The gap ($T_{gap}$) between the operating temperature ($T_{1\ bar}$) for atmospheric pressure delivery and the critical temperature ($T_{cr}$) for zero isotope effect aggravates the interphase isotopic affinity difference. To alleviate the isotope effect during desorption, a local

coordination design strategy striking a balance between thermodynamic stability and isotope effect is established and validated by a series of ZrCo-based alloys. Based on the computational screening results, an optimized Ti-Pd co-doped $Zr_{0.8}Ti_{0.2}Co_{0.8}Pd_{0.2}$ alloy is designed, prepared, and compared with ZrCo, $Zr_{0.8}Ti_{0.2}Co$ and $ZrCo_{0.8}Pd_{0.2}$ alloys. Ti-Pd co-doping enables further suppressed isotope effect than Ti and Pd single doping. Specifically, the $T_{gap}$ of $Zr_{0.8}Ti_{0.2}Co_{0.8}Pd_{0.2}$ alloy reaches 83.99 °C while that of ZrCo, $Zr_{0.8}Ti_{0.2}Co$ and $ZrCo_{0.8}Pd_{0.2}$ alloys are 251.62, 172.00 and 197.41 °C, respectively. Moreover, $Zr_{0.8}Ti_{0.2}Co_{0.8}Pd_{0.2}$ alloy exhibits enhanced cycling stability and a reduction of 30.18% in the maximum of isotopic composition fluctuation during desorption compared to pristine ZrCo alloy. This proposed strategy provides fundamental insights into the hydrogen isotope effect in metal-hydrogen systems and prompts employment of ZrCo-based alloys getter bed for advanced nuclear fusion.

## Results
### Characterization and hydrogen isotope effect of ZrCo alloy
The X-ray diffraction (XRD) refinement and high resolution transmission electron microscopy (HRTEM) verify that the ZrCo alloy with single cubic B2 phase was successfully fabricated for determining accurate isotope effect (Fig. 1a, b). The hydrogen isotope absorption kinetics under atmospheric pressure at room temperature in single- and mixed-isotope systems exhibit no distinct difference in both absorption capacity and rate (Fig. 1c). Specifically, the absorption capacity is near the theoretical capacity (-3 f.u.) and the absorption time until 90% capacity is less than 60 s for both protium and deuterium. Such negligible kinetic isotope effect for absorption benefits rapid isotope storage, resulting from two aspects: (i) the significant driving force originating from the system pressure (-$10^5$ Pa) far beyond the equilibrium plateau pressure at room temperature (-$10^{-3}$ Pa); (ii) the close-packed crystal plane (110) with preferential orientation in the cubic B2 phase, validated by the highest diffraction intensity and dominance of (110) plane in the grain and surface (Fig. 1a, b, d). Moreover, the layered morphology with chemical homogeneity over interlayer and bulk regions was observed (Fig. 1d–f). Generally, an absorption process proceeds, in the metal-hydrogen system, through adsorption, dissociation and diffusion processes. The NEB calculation with zero point energy (ZPE) correction based on the experimental observation and theoretical prediction[55] was thus conducted to elucidate the kinetic isotope effect during absorption (Fig. 1g and Supplementary Table 2). The energy barriers from surface to subsurface and further bulk are 0.76 and 0.57 eV, respectively, which is consistent with previously reported results[55,56]. The energy difference between ZrCo-H/D systems at each step during absorption is less than 0.04 eV, which suggests a slight kinetic isotope effect for absorption. This contributes to further investigation of the kinetic isotope effect for desorption, which means that one can control the isotopic ratio in the solid phase by that in the gas phase.

Differential scanning calorimetry (DSC), temperature programmed desorption (TPD) and thermal desorption spectroscopy (TDS) were used to evaluate the kinetic isotope effect for desorption in single-isotope systems. The presence of a main desorption peak with a shoulder in the DSC and TDS is attributed to the reaction of $ZrCoH_3 \rightarrow ZrCo$ with solute hydrogen release, supported by the XRD patterns at different temperatures (Fig. 1h, i, Supplementary Fig. 2). Additionally, ZrCo deuteride exhibits a lower peak temperature than its hydride in the DSC and TDS, indicating the faster desorption kinetics and weaker thermodynamic stability of deuteride, which accords with the TPD results (Fig. 1j). We further give a quantitative description of the kinetic isotope effect for desorption based on the Kissinger relationship between $\ln \frac{\beta}{T_p^2}$ and $-\frac{1}{RT_p}$ (Eq. (2)). Both the fitting plots depict an obvious linear correlation, of which

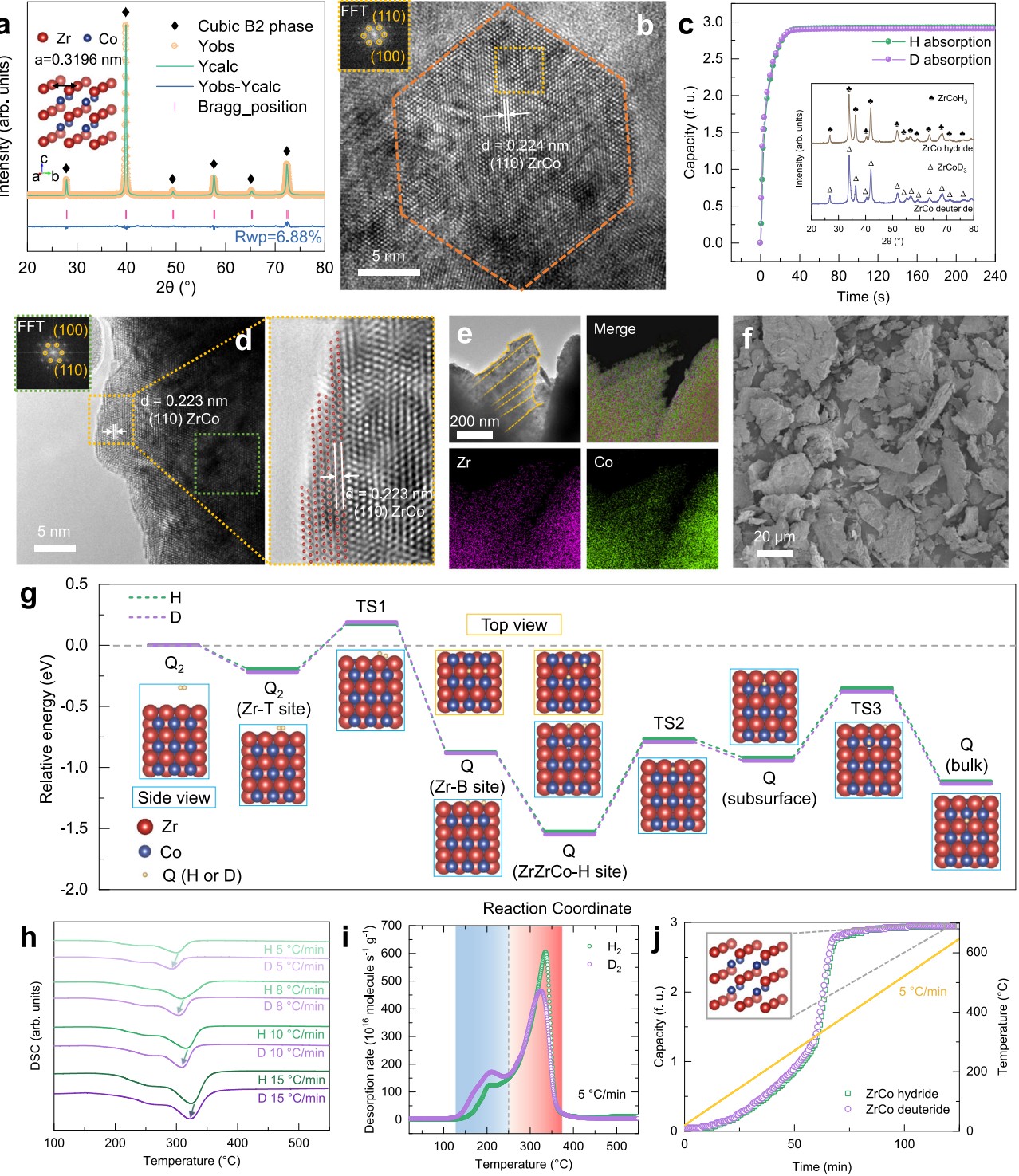

**Fig. 1 | Characterization, kinetic isotopic performance and mechanism of ZrCo alloy. a** XRD pattern and Rietveld refinement result of ZrCo alloy. **b** HRTEM image of a ZrCo grain with the inset graph displaying the FFT image of the region of interest (orange dashed line). **c** Absorption kinetic curves with protium and deuterium, respectively. Inset image shows the XRD patterns of ZrCo hydride and deuteride. **d** HRTEM image of ZrCo particles on the surface. **e** Bright field and corresponding STEM-HAADF image with EDS. **f** SEM image of ZrCo particles. **g** Energy variation of ZrCo-Q (Q = H, D) systems during adsorption, dissociation and diffusion processes with ZrCo (110) plane. T, B and H sites represent top, bridge and hollow sites, respectively. **h** DSC curves of ZrCo hydride and deuteride with a series of heating rates (5, 8, 10, 15 °C/min). **i** TDS curves of ZrCo hydride and deuteride with the heating rate of 5 °C/min. **j** TPD profiles of ZrCo hydride and deuteride with the heating rate of 5 °C/min. In this article, 'f. u.' is adopted as the unit of capacity, which means the mole ratio of isotope and alloy, for isotopic distinction.

the slope represents the apparent activation energy ($E_a$) for desorption (Supplementary Fig. 3). The $E_a$ of ZrCo hydride/deuteride are 111.42 and 97.29 kJ/mol, respectively. The substantial difference between isotopic energy barriers leads to a significant kinetic isotope effect for desorption and limits

accurate isotope supply.

$$\ln \frac{\beta}{T_p^2} = -\frac{E_a}{RT_p} + \frac{AR}{E_a} \qquad (2)$$

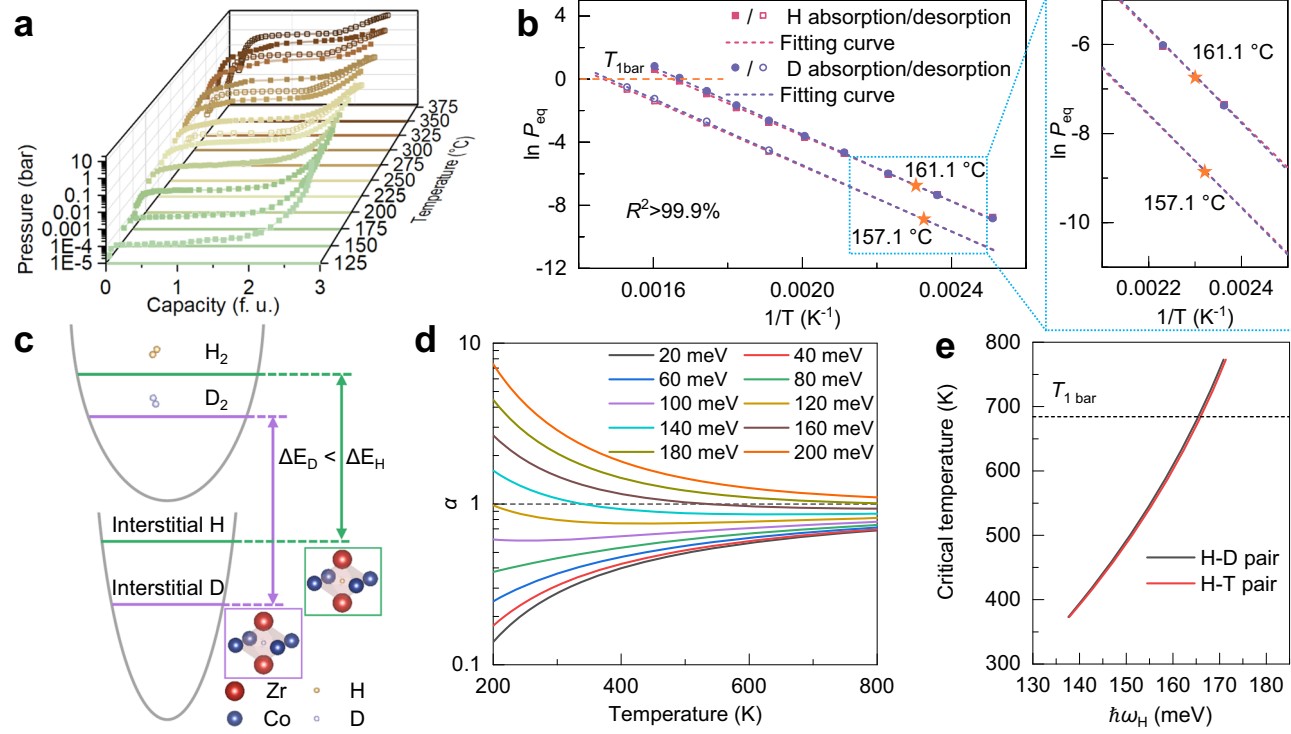

**Fig. 2 | Thermodynamic isotope effect of ZrCo alloy and harmonic oscillator model. a** PCIs of ZrCo alloy for protium absorption and desorption which are represented by open and solid legends, respectively. **b** Van't Hoff plot of ZrCo alloy for protium/deuterium absorption and desorption. **c** Schematic illustration of the potential well curves for molecular and interstitial isotopes. **d** Dependence of separation factor $\alpha$ on the temperature and interstitial vibration energy $\hbar\omega_H$. **e** Relationship between the critical temperature $T_{cr}$ and interstitial vibration energy $\hbar\omega_H$ for binary isotope mixture.

where $\beta$ is the heating rate, $T_p$ represents the peak temperature of DSC curve and $A$ signifies the pre-exponential factor.

To illustrate the difference of thermodynamic stability between hydride and deuteride over a wide temperature range, a series of pressure-composition isotherms (PCIs) were measured (Fig. 2a and Supplementary Fig. 4), of which the plateau pressure ($P_{eq}$) is reckoned as an indicator for thermodynamic stability. The single plateau region is attributed to the one-step reaction, i.e., $2ZrCo + 3Q_2 \leftrightarrow 2ZrCoQ_3$ (Q = H, D), which is in accordance with the DSC and TDS analysis. The temperature dependence of thermodynamic stability was described by Van't Hoff relationship between $\ln P_{eq}$ and $T^{-1}$ (Fig. 2b), which manifests a strong linear correlation ($R^2 > 99.9\%$) and thus a good reliability. Specifically, the isotopic thermodynamic parameters of ZrCo alloy were summarized in Table 1.

### Indicator for the mismatch between thermodynamic stability and isotope effect
It is worth noting that the disparity between the temperature dependences of the thermodynamic stability of ZrCo hydride and

deuteride owing to the isotopic differences of enthalpy and entropy is observed (Fig. 2c). The entropy change ($\Delta S$) operates in the opposite direction compared with the enthalpy change ($\Delta H$) for thermodynamic stability. Moreover, its contribution can be magnified with temperature, leading to the inversion of isotope effect (Fig. 2b), which was also reported in the conventional LaNi$_5$-Q$_2$ (Q = H, D) systems[57]. Similar isotopic inversion phenomena were observed in the low-dimensional materials[58,59]. Herein, an evaluation indicator (critical temperature ($T_{cr}$)), at which hydride and deuteride share the same thermodynamic stability, could be thus readily defined as

$$T_{cr} = \frac{\Delta H_H - \Delta H_D}{\Delta S_H - \Delta S_D} \qquad (3)$$

The $T_{cr}$ for desorption was extrapolated to be 157.11 °C and corresponding plateau pressure is 14 Pa, which is too low to be experimentally evidenced. Inspiringly, the reversible character and large hysteresis between absorption and desorption processes induced by the intrinsic structure-change phase transition allow direct observation of such inversion phenomenon in the absorption process. The difference between plateau pressures for H/D absorption at 150 and 175 °C clearly shows the inversion of thermodynamic isotope effect (Fig. 2b). The accurate $T_{cr}$ for absorption was further calculated to be 161.12 °C, which is very close to the value for desorption. The reversibility of absorption and desorption reactions provides us an effective alternative for experimentally determining isotope effect for forward and reverse reactions.

Delivery at the $T_{cr}$ can enable alleviated isotopic composition fluctuation in favor of the steady-state operation of plasma burning. Not only the isotope effect but also the delivery pressure should be

**Table 1 | Thermodynamic parameters for protium/deuterium absorption and desorption of ZrCo alloy**

| Parameters | Absorption | | Desorption | |
|---|---|---|---|---|
| | Q = H | Q = D | Q = H | Q = D |
| $\Delta H$ (kJ·mol$^{-1}$ Q$_2$) | −85.93 | −88.50 | 86.00 | 87.60 |
| $\Delta S$ (J·K$^{-1}$·mol$^{-1}$ Q$_2$) | −141.93 | −147.85 | 126.15 | 129.87 |
| $T_{1\,bar}$ (°C) | / | / | 408.73 | 401.52 |
| $T_{cr}$ (°C) | 161.12 | | 157.11 | |
| $T_{gap}$ (°C) | / | | 251.62 | |

considered for practical application. To this end, another evaluation indicator (operating temperature ($T_{1\,bar}$)) (Eq. (4)), which represents the temperature for atmospheric pressure delivery, was introduced to characterize the thermodynamic stability of hydride phase.

$$T_{1bar} = \frac{\Delta H}{\Delta S} \qquad (4)$$

In light of the large hysteresis of ZrCo alloy, a relatively high $T_{1\,bar}$ is required (>400 °C), far beyond the corresponding $T_{cr}$ (157.11 °C). In such case, the contribution to thermodynamic stability of entropy change ($\Delta S$) would outweigh that of enthalpy change ($\Delta H$), which leads to the heavy isotope-rich gas during delivery. Furthermore, a comprehensive indicator (temperature gap ($T_{gap}$)), integrating the isotope effect with thermodynamic stability of metal hydride for isotope storage and delivery was proposed (Eq. (5)).

$$T_{gap} = T_{1bar} - T_{cr} \qquad (5)$$

The mismatch between $T_{1\,bar}$ and $T_{cr}$ challenges the compatibility of ZrCo alloy as an isotope storage and delivery material with other components in the tritium plant. The reduction of $T_{gap}$ is expected to mitigate the isotope effect during desorption and improve the composition accuracy of isotope supply.

### Establishment of local coordination design strategy

As a theoretical model referring to the hydrogen atom behavior in the metal-hydrogen system[34,35], the harmonic oscillator model applies the partition functions of molecular and interstitial hydrogen isotopes to deduce the separation factor (Detailed discussion in Supplementary Note 2).

$$\alpha = \frac{Z_{solid}}{Z_{gas}} \qquad (6)$$

where the $Z_{solid}$ and $Z_{gas}$ represent the partition function for hydrogen isotopes in the gas and solid phases, respectively. It is worth noting that the temperature dependence of separation factor is vulnerable to the interstitial vibration energy ($\hbar\omega_H$) (Fig. 2d). In detail, $T_{cr}$ shows a positive correlation with $\hbar\omega_H$, which depends weakly on the pair of isotope mixture (H-D and H-T pairs) (Fig. 2e). This means that the conclusion in the H-D mixed system is hopeful to be extended to the tritium-related system.

Both the elevated $T_{cr}$ and decreased $T_{1\,bar}$ contribute to a narrowed $T_{gap}$ (Fig. 3a). Essentially, the thermodynamic stability and vibration energy of interstices were defined by the coordinated atoms around interstitial hydrogen isotope in the lattice. Herein, the $Zr_{0.75}A_{0.25}Co$ (A=Sc, Ti, V, Y, Zr, Nb, Hf) for Zr site doping and the $ZrCo_{0.75}B_{0.25}$ (B=Cr, Mn, Fe, Co, Ni, Cu, Zn, Mo, Ru, Rh, Pd, Ag, Cd) for Co site doping were constructed and optimized to screen out the most favorable element for diminishing $T_{gap}$ (Fig. 3b and Supplementary Table 3). The decreasing order of $T_{1\,bar}$ follows the decreasing order of doping atom size, which reproduces well the Lundin theory[60]. In addition, the obvious staged distribution of $T_{cr}$ reflects the sensitivity of vibration energy to the local coordination environment of interstitial hydrogen. Ti and Pd were reckoned as the optimal elements to

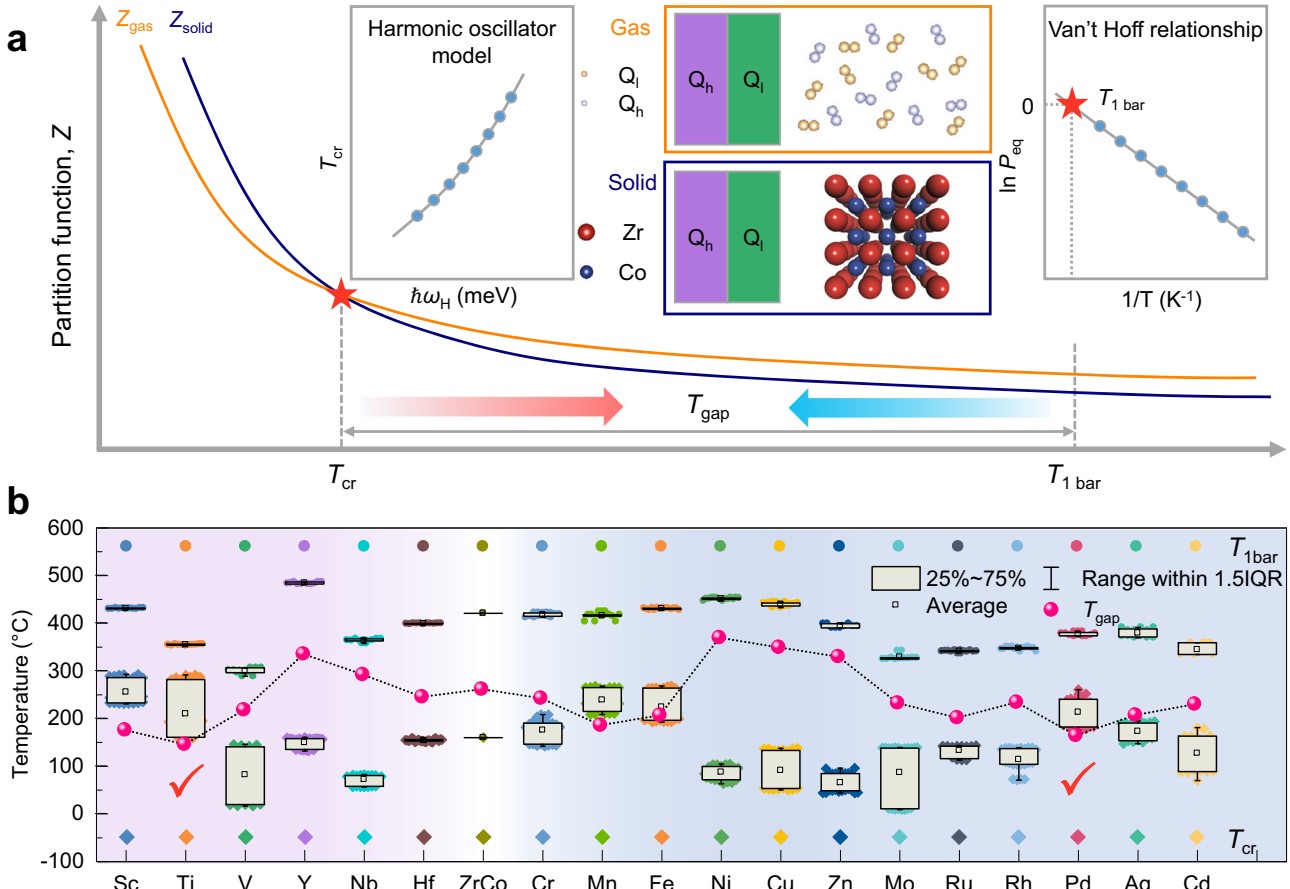

**Fig. 3 | Indicator of $T_{gap}$ and local coordination design strategy. a** Schematic illustration of the $T_{gap}$ between $T_{cr}$ and $T_{1\,bar}$, dictated by the vibration energy and thermodynamic stability of interstitial isotope atoms. **b** Computational screening results of $T_{1\,bar}$, $T_{cr}$ and $T_{gap}$ for a series of element doped ZrCo alloys.

minimize $T_{gap}$ based on the computational screening results. Considering the excellent anti-disproportionation performance, a promising $Zr_{0.8}Ti_{0.2}Co$ alloy was prepared firstly for further investigation.

### Contribution of Ti single doping in the hydrogen isotope absorption and desorption behaviors for $Zr_{0.8}Ti_{0.2}Co$ alloy

The as-cast $Zr_{0.8}Ti_{0.2}Co$ alloy with single cubic B2 phase was prepared (Fig. 4a). In detail, the right shift of diffraction peaks suggests the decreased lattice parameters compared with pristine ZrCo alloy. Moreover, the EDS mapping results reveal the chemically homogeneous nature of $Zr_{0.8}Ti_{0.2}Co$ alloy (Fig. 4b). Additionally, the TEM and HRTEM images give a detailed microstructure information that (110) plane, the close-packed crystal plane of cubic B2 phase, is preferentially presented (Fig. 4c, d). This infers a minor isotope effect during absorption that is supported by the insignificant difference between the diffraction peaks of $Zr_{0.8}Ti_{0.2}Co$ hydride and deuteride (Fig. 4e). To simulate the isotope storage process with the stoichiometry of deuterium and tritium required by the fusion reaction (Eq. (7)), the absorption measurement with equimolar isotope mixture was conducted (Fig. 4f). Furthermore, the protium content in the gas phase during absorption was monitored by gas chromatography. These results demonstrate that $Zr_{0.8}Ti_{0.2}Co$ alloy has a fast absorption

kinetics with a negligible isotope effect comparable as pristine ZrCo alloy.

$$D + T \rightarrow {}^4He + n + 17.6 MeV \qquad (7)$$

PCI and related phase component analysis (Fig. 4g, Supplementary Figs. 5, 6) provide direct evidence of the decisive effect of Ti doping on the thermodynamic behavior for $Zr_{0.8}Ti_{0.2}Co$ alloy: (i) dual plateau for absorption while single plateau for desorption, *i.e.*, absorption-desorption asymmetry, was observed. In addition, the significant difference among $T_{cr}$ for absorption and desorption also suggests disparate reaction pathways; (ii) thermodynamic destabilization of interstitial isotope atoms was realized. In detail, the thermodynamic parameters were calculated by Van't Hoff fitting, as displayed in Table 2. The lower enthalpy change for desorption of $Zr_{0.8}Ti_{0.2}Co$ hydride/deuteride compared with ZrCo hydride/deuteride boosts isotope delivery ability and contributes to a decreased $T_{gap}$.

To elucidate the kinetic pathway that is intimately related to the establishment of thermodynamic equilibrium state, DSC curves were obtained as the desorption kinetics of $Zr_{0.8}Ti_{0.2}Co$ hydride/deuteride. The DSC plots exhibit two endothermic peaks including a weak peak with a relatively low peak temperature and a main peak during

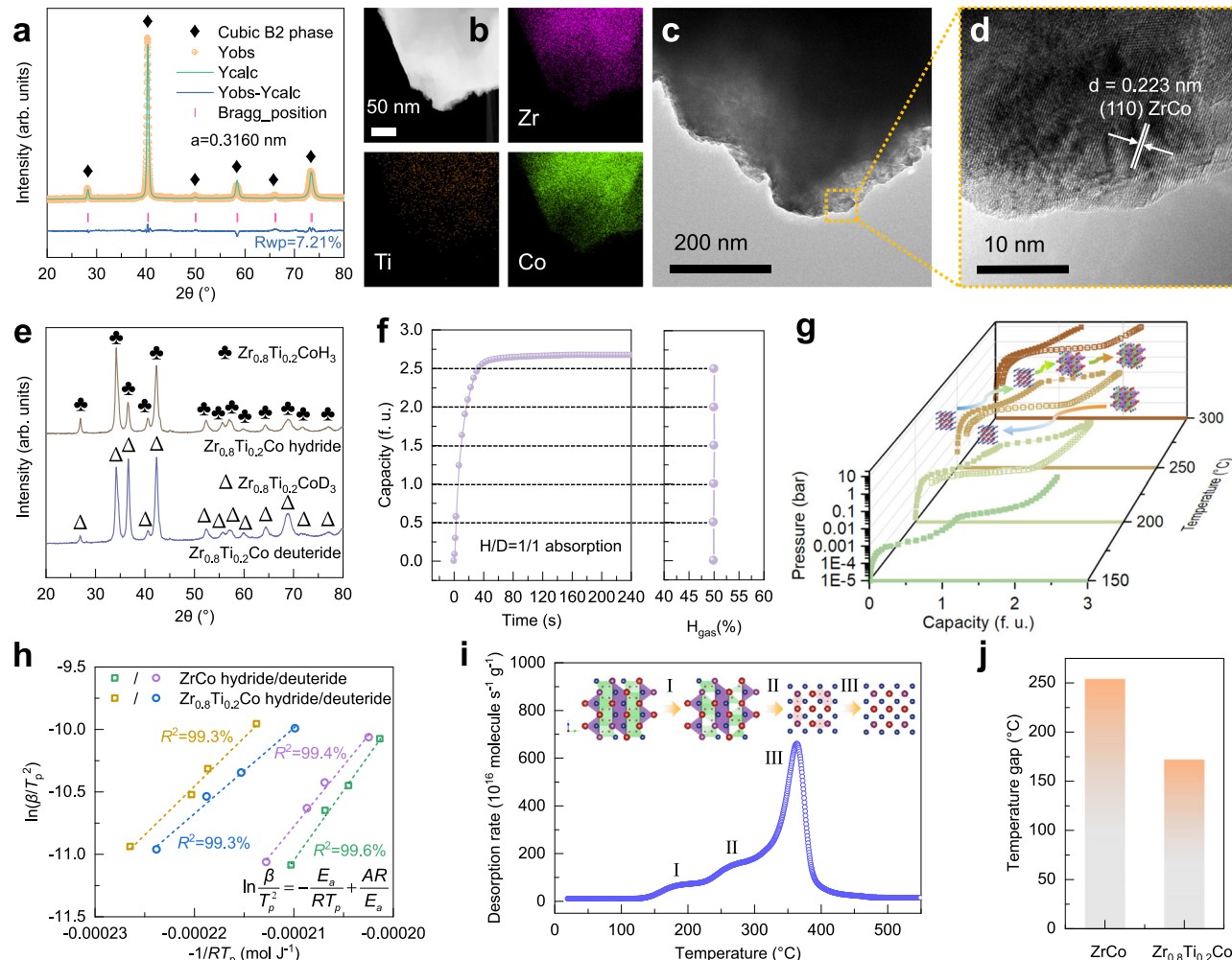

**Fig. 4 | Characterization, isotopic performance and mechanism of $Zr_{0.8}Ti_{0.2}Co$ alloy. a** XRD pattern and Rietveld refinement result, **b,c** TEM and HRTEM images, **d** HAADF-STEM image with EDS of $Zr_{0.8}Ti_{0.2}Co$ alloy. **e** XRD patterns of $Zr_{0.8}Ti_{0.2}Co$ hydride and deuteride. **f** Hydrogen concentration variation in the gas phase during absorption with H/D = 1/1 mixed hydrogen isotopes. **g** PCIs of

$Zr_{0.8}Ti_{0.2}Co$ alloy for protium absorption and desorption which are represented by open and solid legends, respectively. **h** Kissinger fitting curves for desorption of ZrCo and $Zr_{0.8}Ti_{0.2}Co$ hydride/deuteride, respectively. **i** TDS curve of $Zr_{0.8}Ti_{0.2}Co$ hydride. **j** $T_{gap}$ of ZrCo and $Zr_{0.8}Ti_{0.2}Co$ alloys.

**Table 2 | Thermodynamic parameters for protium/deuterium absorption and desorption of $Zr_{0.8}Ti_{0.2}Co$ alloy**

| Parameters | Absorption | | | | Desorption | |
|---|---|---|---|---|---|---|
| | Low plateau | | High plateau | | Q = H | Q = D |
| | Q = H | Q = D | Q = H | Q = D | | |
| $\Delta H$ (kJ·mol$^{-1}$ Q$_2$) | −67.52 | −69.48 | −69.22 | −82.21 | 80.61 | 84.15 |
| $\Delta S$ (J·K$^{-1}$·mol$^{-1}$ Q$_2$) | −106.46 | −113.05 | −144.28 | −166.97 | 131.77 | 139.82 |
| $T_{1\,bar}$ (°C) | / | | / | | 338.75 | 328.85 |
| $T_{cr}$ (°C) | 24.42 | | 299.50 | | 166.75 | |
| $T_{gap}$ (°C) | / | | / | | 172.00 | |

desorption (Supplementary Fig. 7). Owing to the size effect[60], Ti single doping enables $Zr_{0.8}Ti_{0.2}Co$ alloy with a significantly decreased main peak temperature for desorption. Moreover, the reduced slope value of the Kissinger plots for $Zr_{0.8}Ti_{0.2}Co$ alloy indicates a decreased $E_a$ for desorption, which agrees well with the thermodynamic conclusions (Fig. 4h). The $E_a$ of hydride and deuteride are reduced from 111.42 and 97.29 kJ/mol for ZrCo alloy to 77.96 and 68.65 kJ/mol for $Zr_{0.8}Ti_{0.2}Co$ alloy, respectively. The reduced difference between energy barriers helps to mitigate the isotope effect during desorption. To clarify the reason for the asymmetry of absorption and desorption processes, the TDS of $Zr_{0.8}Ti_{0.2}Co$ hydride was measured (Fig. 4i). As per the intensity evolution of gas species in the TDS and the XRD patterns at different temperatures (Supplementary Fig. 8), the desorption process can be divided into three steps: (i) hydrogen release within $\beta$ phase region; (ii) phase transformation from orthorhombic $\beta$ phase to the intermediate hydride phase with cubic structure (B2′ phase); (iii) isostructural phase transition with cubic structure from intermediate hydride phase to cubic B2 phase. The overlap between step (ii) and step (iii) confirms simultaneous hydrogen release corresponding to the phase transformations of $\beta \rightarrow$ B2′ and B2′ $\rightarrow$ B2, further leading to the thermodynamically unstable state for B2′ phase during desorption.

Although the $T_{gap}$ decreased from 251.62 °C for ZrCo alloy to 172.00 °C for $Zr_{0.8}Ti_{0.2}Co$ alloy (Fig. 4j), the reduction is mainly attributed to the decreased $T_{1\,bar}$, associated with the similar electronic configuration and decreased atomic size of Ti compared with Zr. Hence, the interstitial size effect dominates the way for the reduction of $T_{gap}$, but is slightly effective for the improvement of $T_{cr}$. Considering the lowest simulated $T_{gap}$ for Pd among Co site substitution elements and profound isotope effect of Pd metal[61–64], a Ti-Pd co-doped $Zr_{0.8}Ti_{0.2}Co_{0.8}Pd_{0.2}$ alloy was rationally designed to further narrow the $T_{gap}$.

### Synergistic effect of Ti-Pd co-doping on hydrogen isotope effect and cycling stability for $Zr_{0.8}Ti_{0.2}Co_{0.8}Pd_{0.2}$ alloy

$Zr_{0.8}Ti_{0.2}Co_{0.8}Pd_{0.2}$ alloy maintains single cubic B2 phase with homogenous element distribution (Fig. 5a, b). The (110) preferential orientation can also be observed by HRTEM (Fig. 5c). Therefore, the layered alloy intends to keep favorable absorption kinetics with $H_2/D_2 = 1/1$ gas mixture at room temperature as ZrCo and $Zr_{0.8}Ti_{0.2}Co$ alloys (Fig. 5d, Supplementary Fig. 9). The Kissinger fitting results in single-isotope systems indicate further reduction of the energy barrier difference between hydride (93.74 kJ/mol) and deuteride (90.81 kJ/mol), which contributes to a mitigated isotope effect (Supplementary Fig. 10). Additionally, the significant kinetic improvement resulting from Ti-Pd co-doping is verified by TPD measurements for ZrCo, $Zr_{0.8}Ti_{0.2}Co$ and $Zr_{0.8}Ti_{0.2}Co_{0.8}Pd_{0.2}$ hydrides (Supplementary Fig. 11), which ensures high fuel process fluxes. To explore the kinetic isotope effect during desorption in mixed-isotope systems, the TDS profiles of ZrCo, $Zr_{0.8}Ti_{0.2}Co$ and $Zr_{0.8}Ti_{0.2}Co_{0.8}Pd_{0.2}$ alloys saturated in the gas mixture of $H_2/D_2 = 1/1$ were obtained with the heating rate of 5 °C/min. As is evident in Fig. 5e and Supplementary Figs. 12, 13, three gas species including $H_2$, HD and $D_2$ were all detected during the release process.

The consistent onset and peak temperatures among all three gas species signify barely preferential release of any isotopologue. Moreover, HD dominates the TDS among all three species. To quantitatively track the isotopic composition during the whole desorption process, the intensity of $H_2$, HD and $D_2$ species were calibrated. Especially, the HD calibration can be achieved based on the law of hydrogen isotope mass conservation and negligible hydrogen isotope effect of ZrCo-based alloys during absorption. The temperature dependence of the hydrogen concentration of accumulative release gas during desorption was determined. The variation amplitude of isotopic composition in the release gas becomes gradually flat from pristine ZrCo alloy to $Zr_{0.8}Ti_{0.2}Co$ alloy and further $Zr_{0.8}Ti_{0.2}Co_{0.8}Pd_{0.2}$ alloy (Fig. 5f), which reflects the substantial suppression of isotope separation and significant improvement for stable isotope storage and delivery. The detailed data at certain temperatures are list in Supplementary Table 4. Specifically, Ti-Pd co-doping renders a reduction of 30.18% in the maximum of isotopic composition fluctuation during desorption compared with ZrCo alloy, which validates the design protocol. Figure 5g gives a schematic illustration for the desorption process of the hydride phase with mixed hydrogen isotopes. Initially, isotope atoms randomly locate in the lattice interstices of hydride phase. With temperature increasing, they are excited to diffuse from bulk to subsurface and further surface. Combination of isotope atoms occurs on the surface and the molecules with different isotopic modifications are then released into the gas phase[65]. The subsequent isotope exchange process is involved with the interphase isotope exchange between gas and solid phases and homomolecular isotope exchange reaction (HMIE) in the gas phase. In addition, $Zr_{0.8}Ti_{0.2}Co_{0.8}Pd_{0.2}$ alloy shows a discrepancy of thermodynamic pathway between absorption and desorption processes, which is also observed in $Zr_{0.8}Ti_{0.2}Co$ alloy (Fig. 5h). The unapparent plateau region for absorption makes the reaction path elusive. Herein, a series of XRD patterns with different hydrogen contents reveals a two-step reaction mechanism for absorption: (i) formation of the intermediate hydride phase with cubic structure (B2′ phase); (ii) structure-change phase transformation from cubic B2′ phase to orthorhombic ZrCoH$_3$ phase. As for desorption, the single plateau region corresponding to the reaction of orthorhombic ZrCoH$_3$ phase to cubic B2 phase is reconfirmed. Based on the PCIs (Supplementary Figs. 14–17), the corresponding thermodynamic parameters of $Zr_{0.8}Ti_{0.2}Co_{0.8}Pd_{0.2}$ alloy are listed in Supplementary Table 5 and Table 3. Furthermore, the salient driving force caused by the substantial gap between system pressure and equilibrium plateau pressure at room temperature (0.136 Pa at 20 °C) that fitted from Van't Hoff equation ensures the rapid storage kinetics and thus tiny kinetic isotope effect during absorption. For desorption, the minor difference of $T_{1\,bar}$ between $Zr_{0.8}Ti_{0.2}Co_{0.8}Pd_{0.2}$ and $Zr_{0.8}Ti_{0.2}Co$ alloys specifies the little influence of Pd substitution for Co on the stability of hydride phase. It should be noted that the mitigated $T_{gap}$ is mainly attributed to the increase of $T_{cr}$.

To understand the explicit contribution of Pd single doping for hydrogen isotope effect, $ZrCo_{0.8}Pd_{0.2}$ alloy was prepared. The single B2 phase, hierarchical layered microstructure at the micro/nanometer scales and preferred orientation of (110) plane are also demonstrated by XRD, SEM and TEM (Supplementary Figs. 18 and 19a, b). Additionally, the STEM-HAADF image and corresponding EDS mapping demonstrate the homogeneous distribution and accurate content of alloy elements (Supplementary Fig. 19c, d). The thermodynamic parameters of $ZrCo_{0.8}Pd_{0.2}$ alloy were obtained based on a series of PCIs (Supplementary Figs. 20–23) and listed in Supplementary Table 6. The single plateau region and nearly consistent $T_{cr}$ for both absorption and desorption processes confirm the reversibility of phase transition and the reliability to deduce isotopic behaviors from reversible reactions. The $T_{gap}$ for $ZrCo_{0.8}Pd_{0.2}$ alloy is 197.41 °C, of which the reduction is mainly contributed to the increase of $T_{cr}$ from 157.11 °C for ZrCo alloy to 193.31 °C for $ZrCo_{0.8}Pd_{0.2}$ alloy.

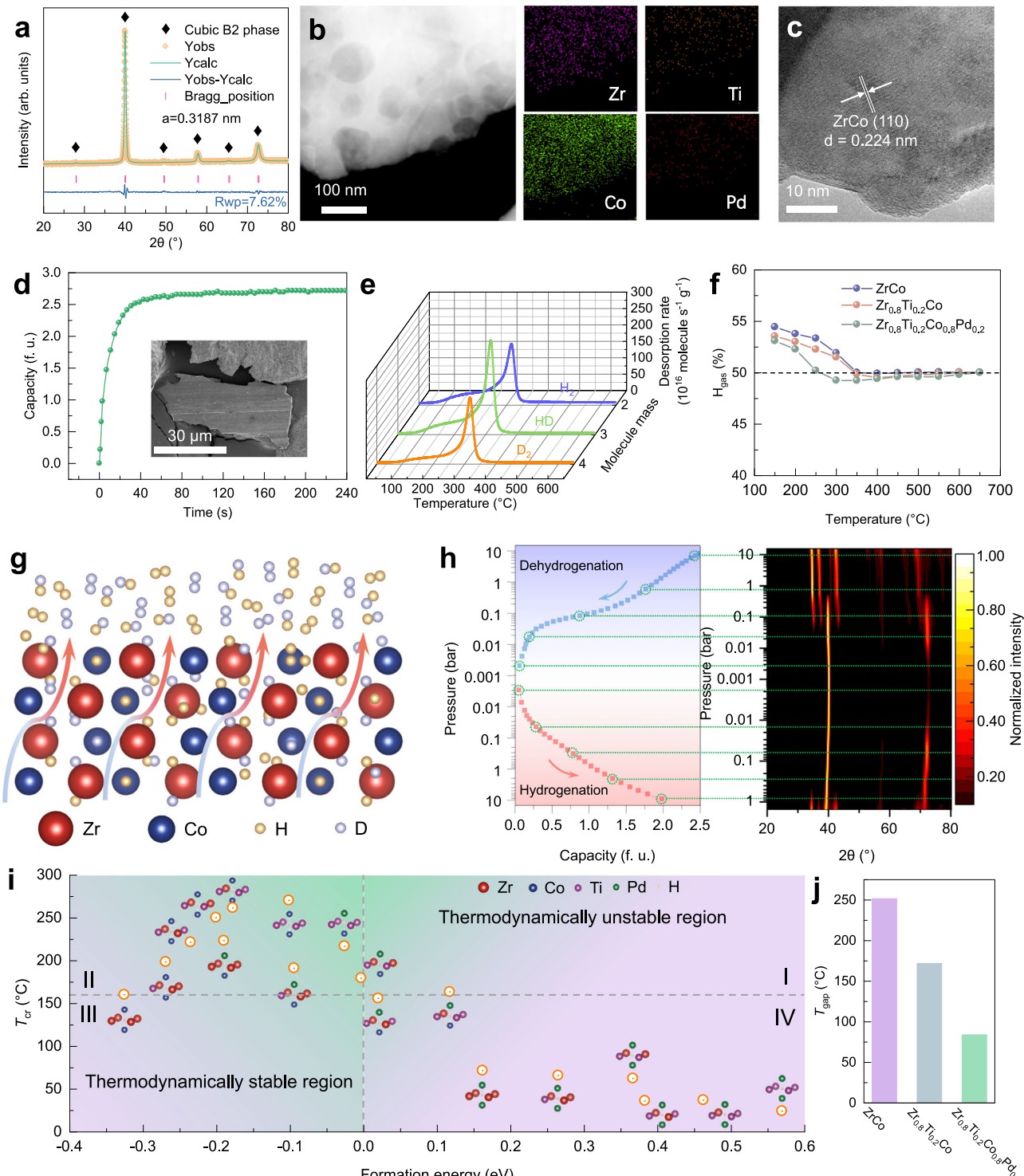

**Fig. 5 | Characterization, isotopic performance and mechanism of Zr$_{0.8}$Ti$_{0.2}$Co$_{0.8}$Pd$_{0.2}$ alloy. a** XRD pattern and Rietveld refinement result, (**b**) HRTEM image, (**c**) HAADF-STEM image with EDS of Zr$_{0.8}$Ti$_{0.2}$Co$_{0.8}$Pd$_{0.2}$ alloy. **d** Typical hydrogenation kinetic behavior of Zr$_{0.8}$Ti$_{0.2}$Co$_{0.8}$Pd$_{0.2}$ alloy with layered microstructure which can be observed from the inset SEM image. **e** TDS of Zr$_{0.8}$Ti$_{0.2}$Co$_{0.8}$Pd$_{0.2}$ sample saturated in the mixed gas with H$_2$/D$_2$ = 1/1. **f** Cumulative

hydrogen concentration of release gas during the whole desorption process. **g** Schematic illustration of the desorption process of the hydride phase with mixed hydrogen isotopes. **h** Phase identification along the de-/hydrogenation processes of Zr$_{0.8}$Ti$_{0.2}$Co$_{0.8}$Pd$_{0.2}$ alloy. **i** Correlation between $T_{cr}$ and corresponding formation energy of interstitial isotopes in Zr$_{0.8}$Ti$_{0.2}$Co$_{0.8}$Pd$_{0.2}$ alloy. **j** $T_{gap}$ of ZrCo, Zr$_{0.8}$Ti$_{0.2}$Co and Zr$_{0.8}$Ti$_{0.2}$Co$_{0.8}$Pd$_{0.2}$ alloys.

To elaborate the interaction between the interstitial hydrogen atom and Zr$_{0.8}$Ti$_{0.2}$Co$_{0.8}$Pd$_{0.2}$ alloy matrix, all possible interstices were constructed and optimized. According to the harmonic oscillator model and density functional theory (DFT) calculations, there is a close correlation between the formation energy and $T_{cr}$ for interstitial

hydrogen atoms (Fig. 5i). All interstices with negative formation energy exhibit improved $T_{cr}$ (Zone II) whereas those with positive formation energy are expected to have decreased $T_{cr}$ (Zone IV). This means all thermodynamically stable interstices can assist in enhancing the $T_{cr}$ of Zr$_{0.8}$Ti$_{0.2}$Co$_{0.8}$Pd$_{0.2}$ alloy. More importantly,

**Table 3 | Thermodynamic parameters for protium/deuterium absorption and desorption of $Zr_{0.8}Ti_{0.2}Co_{0.8}Pd_{0.2}$ alloy**

| Parameters | Desorption | |
|---|---|---|
| | Q = H | Q = D |
| $\Delta H$ (kJ·mol$^{-1}$ Q$_2$) | 68.66 | 82.97 |
| $\Delta S$ (J·K$^{-1}$·mol$^{-1}$ Q$_2$) | 109.89 | 136.35 |
| $T_{1\,bar}$ (°C) | 351.81 | 335.51 |
| $T_{cr}$ (°C) | 267.82 | |
| $T_{gap}$ (°C) | 83.99 | |

Ti-Pd co-doping synergistically induces vibration enhancement, *i.e.*, increased $T_{cr}$, as well as thermodynamic destabilization, *i.e.*, decreased $T_{1\,bar}$, within the thermodynamically stable zone (Zone II), which remarkably narrowed the $T_{gap}$ from 251.62 °C (ZrCo alloy) to 83.99 °C ($Zr_{0.8}Ti_{0.2}Co_{0.8}Pd_{0.2}$ alloy) (Fig. 5j).

Furthermore, the cycling stability of isotope storage and delivery materials proves critical for applicability in the tritium-related field. Targeting the tritium self-sufficiency and operational safety for fusion reactors, the design of tritium plant should minimize the tritium start-up inventory and maintain the limited reserve tritium inventory. The SDS is required to provide stable cycling processes to meet these stringent requirements. Thus, we further evaluated the effect of Ti-Pd co-doping on the cycling stability of $Zr_{0.8}Ti_{0.2}Co_{0.8}Pd_{0.2}$ alloy with deuterium, where the specific process parameters are shown in Supplementary Fig. 24.

As demonstrated in Supplementary Fig. 25, the capacity retention and stable cyclic capacity of $Zr_{0.8}Ti_{0.2}Co_{0.8}Pd_{0.2}$ alloy after 50 cycles are 61.94% and 1.66 f. u., over twofold higher than 25.91% and 0.715 f. u. of ZrCo alloy, which emphasizes the cooperative enhancement of Ti-Pd co-doping to retard disproportionation and prolong cycling stability by thermodynamic destabilization. The cycling capacity attenuation of $Zr_{0.8}Ti_{0.2}Co_{0.8}Pd_{0.2}$ alloy arises from two aspects: (i) the hydrogen-induced disproportionation reaction (HID); (ii) the iso-structural phase transition with cubic structure induced by spatial confinement of in situ disproportionation products ($ZrCo_2$ and $ZrH_2$). The contradictory results between appreciable capacity fading and trace amount of disproportionation products in the XRD profiles (Supplementary Fig. 25c, d) confirm the much larger contribution of the latter to capacity attenuation, which is favorable to the minimization of tritium inventory and retention.

In summary, through comprehensive experimental and theoretical analysis, we obtain accurate isotopic performance and elucidate corresponding mechanism for ZrCo alloy. To alleviate the isotopic composition deviation governed by the kinetic isotope effect during desorption, we propose a quantitative indicator of $T_{gap}$ to describe the mismatch between thermodynamic stability and isotope effect, and further establish a local coordination design strategy coupling thermodynamic destabilization with vibration enhancement of interstitial isotopes for rational isotope engineering. A minimized $T_{gap}$ is desired to realize precise isotope supply to ensure safe and efficient operation of plasma burning. Based on the computational screening results, Ti and Pd elements are screened out, and an optimized Ti-Pd co-doped $Zr_{0.8}Ti_{0.2}Co_{0.8}Pd_{0.2}$ alloy is designed and prepared. Ti-Pd co-doping endows $Zr_{0.8}Ti_{0.2}Co_{0.8}Pd_{0.2}$ alloy a one-third lower isotopic composition deviation during desorption and an over two-fold higher cycling capacity compared with pristine ZrCo alloy. The explicit contribution of Ti/Pd single doping and synergistic effect of Ti–Pd co-doping on the hydrogen isotope effect are elaborated. The impressive success in suppressing hydrogen isotope effect validates the design principle for the isotope engineering of ZrCo-based alloys, thus prompting the advancement of isotope storage and delivery material in nuclear fusion.

## Methods

### Sample preparation
Zirconium (99.9%) and cobalt (99.9%) were supplied by General Research Institute for Non-ferrous Metals. Titanium (99.9%) was purchased from Alfa Aesar. Palladium (99.95%) was purchased from Beijing Dream Material Technology Co., Ltd. As a typical synthesis of alloy, ZrCo, $Zr_{0.8}Ti_{0.2}Co$, $ZrCo_{0.8}Pd_{0.2}$ and $Zr_{0.8}Ti_{0.2}Co_{0.8}Pd_{0.2}$ alloys were prepared by induction levitation melting method. High purity metals based on the chemical composition were melted together followed by remelting process for four times in a water-cooled copper crucible to ensure chemical homogeneity. During melting process, the melting chamber was full of argon atmosphere of 1.4 bar to protect the sample from oxidation.

### Characterizations
X-ray diffraction (XRD) patterns were obtained on an X-ray powder diffractometer (Smartlab, Rigaku) with Cu K$_\alpha$ radiation ($\lambda = 1.5406$ Å). *FullProf Suite* program was employed to determine lattice constants by Rietveld refinement. The microstructure was observed by scanning electron microscopy (SEM, Hitachi SU-70) and transmission electron microscopy (TEM, Tecnai G2 F20 S-TWIN) equipped with energy dispersive X-ray spectrometer (EDS, Oxford X-MAX 80 T).

### Hydrogen isotope effect measurements
The as-cast samples were activated at 500 °C under vacuum condition after being polished by grinding wheel. Further, 5 bar pure hydrogen was loaded at 100 °C and maintained for several hours until the activation process was completed. The activated samples were preserved in an argon-filled glove box with $O_2$ and $H_2O$ level less than 0.01 ppm for subsequent experiments. The PCIs (Pressure-composition isotherms) were obtained by a series of gas-solid equilibrium under different hydrogen pressures at a setting temperature in the self-made Sievert's type platform (leak rate<$1 \times 10^{-9}$ cc/sec, Agilent). The hydrogen pressure was measured by vacuum gauges (1 Torr and 1000 Torr, INFICON) and pressure transmitter (10 bar ABS, WIKA). Hydrogen isotope absorption/desorption kinetics were recorded by a VGC503 (INFICON). The gas chromatograph (GC, Agilent 7890B) equipped with a molecular sieve column (5 A) and thermal conductivity detector (TCD) was employed to monitor the gas composition during absorption. Differential scanning calorimetry (DSC) was conducted on a simultaneous thermal analyzer (NETZSCH, STA449 F3 Jupiter) from 20 to 550 °C. Temperature programmed desorption (TPD) curves were obtained from room temperature to setting temperature with the heating rate of 5 °C/min. Thermal desorption spectroscopy (TDS) using a quadrupole mass spectrometer (QMS) was adopted to evaluate isotopic selectivity from 20 to 650 °C with the heating rate of 5 °C/min. Sample powder with certain isotopic composition was placed in the chamber and pre-evacuated by turbomolecular pump. The desorption signals corresponding to isotopically pure gas components were calibrated by standard gases. Absorption and desorption cycle tests consist of deuterium absorption under atmospheric pressure at room temperature and temperature programmed desorption with deuterium from room temperature to 380 °C.

### Theoretical calculations
The first-principle calculations were conducted using density functional theory (DFT) based on the Vienna Ab Initio Simulation Package (VASP). The electronic exchange-correlation interaction was described by the projected augmented wave (PAW) method and generalized gradient approximation (GGA) together with functional of Perdew–Burke–Ernzerhof (PBE). A kinetic energy cutoff of 400 eV was applied for theoretical simulations. The convergence criterion and ionic relaxation criterion were set to $1 \times 10^{-6}$ eV and 0.01 eV Å$^{-1}$, respectively.

For the simulation of isotope absorption process, the $3 \times 2$ supercell with ZrCo (110) plane containing a slab of six atomic layers (top three relaxed, bottom three fixed) and 15 Å vertical vacuum space was established. The climbing image nudging elastic band (CI-NEB) method was employed to track the energy variation during adsorption, dissociation and diffusion processes for both protium and deuterium. The zero-point energy (ZPE) correction was carried out to determine the formation energy and energy barrier.

For doping elements screening, $Zr_{0.75}A_{0.25}Co$ (A=Sc, Ti, V, Y, Zr, Nb, Hf) for Zr site doping and $ZrCo_{0.75}B_{0.25}$ (B=Cr, Mn, Fe, Co, Ni, Cu, Zn, Mo, Ru, Rh, Pd, Ag, Cd) for Co site doping with supercell of $2 \times 2 \times 2$ for cubic phase were constructed. A k-point mesh of $4 \times 4 \times 4$ was used for Brillouin zone sampling. The calculation of vibration frequency for interstitial isotope was conducted with the ground state structures after geometry optimization. For exploring the vibration energy of interstitial isotope for all possible interstices in $Zr_{0.8}Ti_{0.2}Co_{0.8}Pd_{0.2}$ alloy, supercell of $3 \times 3 \times 3$ for cubic phase was employed. The k-point meshes of $4 \times 4 \times 4$ was used for Brillouin zone sampling. Crystal structures were built using Visualization for Electronic and Structure Analysis (VESTA) software.

## Data availability
The source data that support the findings of this study are available from the corresponding author upon request.

## Code availability
The codes that support the findings of this study are available from the corresponding author upon request.

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

## Acknowledgements

We are grateful for the financial supports from the National Natural Science Foundation of China (52071286 and U2030208) and the National Key Research and Development Program of China (2022YFE03170002). We thank Dr. Zhemin Wu for the HRTEM imaging and analyzing.

## Author contributions

J.Q. and X.H. contributed equally to this work. J.Q., X.H., X.X., H.K. and L.C. conceived the idea and designed the experiments. J.Q. and X.Z. conducted materials syntheses, characterizations, and measurements. P.Z., S.Z. and R.L. conducted the theoretical calculation. X.H., F.J., Y.Y., J.S., X.F., Y.S. and W.L. helped the isotope effect test. All authors discussed and analysed the data. J.Q. wrote the draft manuscript. J.Q., X.H., X.X., H.K. and L.C. revised the manuscript.

## Competing interests

The authors declare no competing interests.

## Additional information

**Supplementary information** The online version contains Supplementary Material available at https://doi.org/10.1038/s41467-024-47250-3.

