## [Peer Review File · Nature Communications]

Isotope engineering achieved by local coordination design in Ti-Pd co-doped ZrCo-based alloysREVIEWER COMMENTS

Reviewer #1 (Remarks to the Author):

The work describes the development of Zr-Co alloys for hydrogen isotopes storage. The theme is of interest for fusion fuel cycle applications and, more in general, for applications related to the hydrogen technologies thought for supporting the energy transition.

The work methodology adopted by the authors relies on an effective thermodynamic and kinetic analysis whose results are verified by experiments articulated in a sound manner.

The following comments are suggested in order to improve the quality of the paper.

The cycling stability of the Zr-Co alloy is an important aspect that could limit the applicability of this alloy as mentioned in the Introduction by the authors that cite the refs. 15 and 21-23. The work clearly demonstrates the effect of Ti-Pd co-doping in reducing the T_{gap} while no comment is reported about the capability of these elements (Ti and Pd) of affecting the cycling stability of the Zr-Co alloy. Could the authors argue on this aspect?

Although the T_{gap} reduction can in general improve the performances of the Zr-Co alloy in storing and delivery the hydrogen isotopes in a fusion system, such an effect could be better evaluated when specific process requirements of the fusion fuel cycle would be defined (e.g., D-T delivery times, flow rates, etc.). For instance, in case D-T can be provided in certain (long) times or the D/T ratio can be rebalanced through different systems (e.g., additional storing and then delivery of small amounts of pure D and T) the advantage to have the prompt availability of 50/50 D-T mixtures could be less important than other alloy characteristics, such the above-mentioned cycling stability. Any discussion along these lines could further strengthen the scope of the work.

Finally, a list of acronyms could make the paper easier to read.

Reviewer #2 (Remarks to the Author):

The manuscript describes the importance of thermodynamic and kinetic isotope effect on Ti-Pd substituted ZrCo alloy system in general and the $Zr_{0.8}Ti_{0.2}Co_{0.8}Pd_{0.2}$ alloy in particular. Various types of experimental methods are employed to examine both the effects. Further, the screening of Ti and Pd alloy substitution was carried out on the basis of computational method. The results are promising to reduce the temperature gap (T_{gap}) between the critical temperature (T_{cr}) and the operational temperature of hydrogen isotope delivery (T 1 bar).

However, these studies are not new in entirety. The concepts of development of hydrogen isotope storage and delivery system based on ZrCo alloy systems are well documented. Hence, this manuscript does not provide any new insight into the science and technology directional development in this field.

Hence, it may not be suitable for publication in Nature Communications.

Reviewer #3 (Remarks to the Author):

The prominent isotope effect of ZrCo alloy originating from the most significant mass difference among hydrogen isotopes induces the isotope composition deviation in the fusion fuel gas. The manuscript presents comprehensive modeling and extensive characterization analysis to demonstrate the intrinsic reasons for the difference in the isotope effects between hydrogen isotope absorption and desorption processes. Authors recognize that the kinetic hydrogen isotope effect during the desorption process dominates the isotope composition fluctuation for ZrCo alloy. The manuscript presents a local coordination design strategy for minimizing the hydrogen isotope effect of ZrCo-based alloys during the desorption process. Therefore, they propose the novel Ti-Pd co-doped $Zr_{0.8}Ti_{0.2}Co_{0.8}Pd_{0.2}$ alloy that achieves more consistent isotopic behavior through thermodynamic destabilization with vibration enhancement of interstitial isotopes. As a result, it demonstrates that the alloys developed following

this principle show minor isotope composition deviation during the desorption process. The manuscript is clearly written, well organized. As such, I believe this manuscript stands out from many other manuscripts in this area and can be recommended for publication after clarifying the questions and concerns raised below:

1. The authors point to the importance of accurate thermodynamic parameters with regard to the determination of thermodynamic isotope effect, especially for the separation factor (α) and further critical temperature (T_{cr}) for zero isotope effect. However, the equilibrium plateau pressures of PCI curves get sloped with the increase of the amount and type of substitution elements (Fig. S13-S16). How the authors define the equilibrium plateau pressure with a strong sloped curve?
2. L126-129, "Such kinetic isotope effect for the absorption process benefits rapid isotope storage, resulting from two aspects: (i) the significant driving force originating from the system pressure (~ 105 Pa) far beyond the equilibrium plateau pressure (10^{-3} Pa)", the negligible kinetic isotope effect of the designed ZrCo-based alloys during the absorption process should be further demonstrated by calculating the equilibrium plateau pressure at room temperature (20 °C).
3. Explain how to quantitatively determine the isotope ratio during the whole desorption process according to the TDS curves, especially for the HD calibration. And please provide a table including the detailed data of Fig. 5f.
4. Measure the thermodynamic parameters of ZrCo_{0.8}Pd_{0.2} alloy to understand the explicit contribution of single Pd element doping.

Point-by-point Response to the Reviewers' Comments

Nature Communications manuscript NCOMMS-23-54158A

Title: Isotope engineering achieved by local coordination design in Ti-Pd co-doped ZrCo-based alloys

We very much appreciate your helpful comments and suggestions, especially regarding points of confusion in the manuscript. Our responses to reviewer comments are shown in blue. In the revised manuscript and supporting information, changes are highlighted in yellow and also included in the response to the comment that motivated the change.

Reviewer #1

The work describes the development of Zr-Co alloys for hydrogen isotopes storage. The theme is of interest for fusion fuel cycle applications and, more in general, for applications related to the hydrogen technologies thought for supporting the energy transition.

The work methodology adopted by the authors relies on an effective thermodynamic and kinetic analysis whose results are verified by experiments articulated in a sound manner.

The following comments are suggested in order to improve the quality of the paper.

Reply: Thanks for the reviewer's positive comment on our work.

1. The cycling stability of the Zr-Co alloy is an important aspect that could limit the applicability of this alloy as mentioned in the Introduction by the authors that cite the refs. 15 and 21-23. The work clearly demonstrates the effect of Ti-Pd co-doping in reducing the T_{gap} while no comment is reported about the capability of these elements (Ti and Pd) of affecting the cycling stability of the Zr-Co alloy. Could the authors argue on this aspect?

Reply: Thanks for the reviewer's constructive suggestion, which improved the quality of our work.

To ensure the tritium self-sufficiency and operational safety of fusion reactors, the design of tritium plant should minimize the tritium start-up inventory and maintain a limited reserve tritium inventory. The storage and delivery system (SDS) is required to provide

stable cycling processes to meet these stringent requirements [*International Journal of Hydrogen Energy* 46 (2021)]. Thus, we have evaluated the effect of Ti-Pd co-doping on the cycling stability of Zr-Co alloy with deuterium. Since hydrogen pressures and temperatures are indispensable for the hydrogen-induced disproportionation reaction (HID), we choose 380 °C as the delivery temperature. The relatively low cycling delivery temperature can avoid severe disproportionation under hydrogen back pressures.

In the revised manuscript, we added:

Page 2: Compared to the pristine ZrCo alloy, Ti-Pd co-doping in the $Zr_{0.8}Ti_{0.2}Co_{0.8}Pd_{0.2}$ alloy enables consistent isotope desorption kinetics together with a three-fold lower T_{gap} , a five-fold lower kinetic barrier difference and enhanced cycling stability.

Page 7: Moreover, the $Zr_{0.8}Ti_{0.2}Co_{0.8}Pd_{0.2}$ alloy exhibits enhanced cycling stability and a reduction of 30.18% in the maximum of isotope composition fluctuation during the desorption process compared to the pristine ZrCo alloy.

Page 23: Additionally, the cycling stability of isotope storage and delivery materials proves critical for applicability in the tritium-related field. Targeting the tritium self-sufficiency and operational safety for fusion reactors, the design of tritium plant should minimize the tritium start-up inventory and maintain a limited reserve tritium inventory. The SDS is required to provide stable cycling processes to meet these stringent requirements. Thus, we further evaluated the effect of Ti-Pd co-doping on the cycling stability of the ZrCo alloy with deuterium, where the specific process parameters are shown in Fig. S24.

Page 23: As demonstrated in Fig. S25, the capacity retention and stable cyclic capacity of the $Zr_{0.8}Ti_{0.2}Co_{0.8}Pd_{0.2}$ alloy after 50 cycles are 61.94% and 1.66 f. u., over 2-fold higher than the 25.91% and 0.715 f. u. of the ZrCo alloy, which emphasizes the cooperative enhancement of Ti-Pd co-doping to retard disproportionation and prolong cycling stability by thermodynamic destabilization. The cycling capacity attenuation of the $Zr_{0.8}Ti_{0.2}Co_{0.8}Pd_{0.2}$ alloy arises from two aspects: (i) the hydrogen-induced disproportionation reaction (HID); (ii) the isostructural phase transition with cubic structure induced by spatial confinement by the *in situ* disproportionation products ($ZrCo_2$

and ZrH₂). The contradictory results between appreciable capacity fading and trace amount of disproportionation products in the XRD profiles (Fig. S25c, d) confirmed the much larger contribution of the latter to capacity attenuation, which is favorable to the minimization of tritium inventory and retention.

In the revised supporting information, we added:

Page 3: Absorption and desorption cycle tests consist of deuterium absorption under atmospheric pressure at room temperature and temperature programmed desorption with deuterium from room temperature to 380 °C.

Fig. S24 Cycling process parameters of system pressures (a) and corresponding temperatures (b) for the absorption and desorption processes.

Fig. S25 Cyclic deuterium absorption (a) and desorption (b) capacities of ZrCo, $Zr_{0.8}Ti_{0.2}Co$ and $Zr_{0.8}Ti_{0.2}Co_{0.8}Pd_{0.2}$, and corresponding XRD patterns of their deuterides (c) and alloys (d) at 50th cycle.

2. Although the T_{gap} reduction can in general improve the performances of the Zr-Co alloy in storing and delivery the hydrogen isotopes in a fusion system, such an effect could be better evaluated when specific process requirements of the fusion fuel cycle would be defined (e.g., D-T delivery times, flow rates, etc.). For instance, in case D-T can be provided in certain (long) times or the D/T ratio can be rebalanced through different systems (e.g., additional storing and then delivery of small amounts of pure D and T) the advantage to have the prompt availability of 50/50 D-T mixtures could be less important than other alloy characteristics, such the above-mentioned cycling stability. Any discussion along these lines could further strengthen the scope of the work.

Reply: Thanks for the reviewer's great suggestion.

To obtain high-purity and required DT ratio from exhausted fluxes to fuel the plasma, the inner fuel cycle (IFC) of a fusion reactor includes a tokamak exhaust processing system (TEP), an isotope separation system (ISS) and a storage and delivery system (SDS). Owing to the design requirements of a fusion reactor for rapid fueling and accurate isotope composition, major exhaust gases are processed through the TEP and SDS while the remaining gases are allowed to transfer to the ISS for separation of deuterium and tritium [*Fusion Engineering and Design* 159 (2020)]. It is highly desirable to directly obtain the delivery stream from the SDS with stable isotope composition whereas the available rebalance of isotope composition can only be used as a supplementary method [*Fusion Engineering and Design* 192 (2023)]. For safety and efficiency concerns, metal hydride technology as a feasible solution for the SDS allows compact tritium storage at low pressures, tritium delivery at moderate temperatures, automatic control, and remote operation. Thus, the prompt availability to the delivery stream with stable isotope composition in metal hydrogen systems is critical to improve the stability and efficiency of plasma operation.

The discussions of cycling stability have already been presented. On the other hand, we have compared the delivery rates for the ZrCo, Zr_{0.8}Ti_{0.2}Co and Zr_{0.8}Ti_{0.2}Co_{0.8}Pd_{0.2} hydrides. The kinetic improvement resulting from the thermodynamic destabilization and vibration enhancement via Ti-Pd co-doping is significant, which ensures high fuel process fluxes.

Macroscopic performances of the SDS including storage and delivery rates, durable times and isotope composition variations of fuel stream are depended on the metal hydride getter bed which is the main component of the SDS. Notably, the coordination environment of interstitial hydrogen isotopes in the metal hydride should be the most crucial basis of isotope system, which determines the interphase isotope affinity and distribution, further the storage and delivery performances. Thus, developing an effective method that bridges from interstitial coordination environment at atomic level to scale-up isotope behaviors is highly urgent and necessary, which could help a general design strategy for advancing isotope storage and delivery materials. In this work, we proposed

the concept of T_{gap} and bottom-up design strategy that comprises thermodynamic destabilization with vibration enhancement of interstitial isotopes for rational isotope engineering. Ti and Pd elements were screened out by our design guideline. Based on the effective thermodynamic and kinetic analysis, Ti-Pd doping in the $\text{Zr}_{0.8}\text{Ti}_{0.2}\text{Co}_{0.8}\text{Pd}_{0.2}$ alloy not only maintains the prompt hydrogen isotope absorption and desorption kinetics but also suppresses the isotope composition deviation and enables a trivial isotope effect. Moreover, our added discussions about cycling stability indicate that the $\text{Zr}_{0.8}\text{Ti}_{0.2}\text{Co}_{0.8}\text{Pd}_{0.2}$ alloy exhibits enhanced cycling stability.

In the revised manuscript, we added:

Page 4: To obtain high-purity and required DT ratio from exhausted fluxes to fuel the plasma, the inner fuel cycle (IFC) of a fusion reactor includes a tokamak exhaust processing system (TEP), an isotope separation system (ISS) and a storage and delivery system (SDS). Owing to the design requirements of a fusion reactor for rapid fueling and accurate isotope composition, major exhaust gases are processed through the TEP and SDS while the remaining gases are allowed to transfer to the ISS for separation of deuterium and tritium.

Page 20: Additionally, the significant kinetic improvement resulting from Ti-Pd co-doping is verified by TPD measurements for the ZrCo , $\text{Zr}_{0.8}\text{Ti}_{0.2}\text{Co}$ and $\text{Zr}_{0.8}\text{Ti}_{0.2}\text{Co}_{0.8}\text{Pd}_{0.2}$ hydrides (Fig. S11), which ensures high fuel process fluxes.

In the revised supporting information, we added:

Fig. S11 TPD profiles of ZrCo , $\text{Zr}_{0.8}\text{Ti}_{0.2}\text{Co}$ and $\text{Zr}_{0.8}\text{Ti}_{0.2}\text{Co}_{0.8}\text{Pd}_{0.2}$ hydrides with the heating rate of

5 °C/min.

3. Finally, a list of acronyms could make the paper easier to read.

Reply: Thanks for the reviewer's great comment.

We now consistently include detailed description of the notation at their first instance in the text to clarify what each symbol represents and summarize all these information into a list of acronyms.

In the revised manuscript, we added:

Page 4: A list of abbreviations and their corresponding detailed description mentioned in this paper is presented in the Table S1 to improve the readability.

Page 10: where β is the heating rate, T_p represents the peak temperature of DSC curve and A signifies the pre-exponential factor.

In the revised supporting information, we added:

Table S1 List of abbreviations and their corresponding detailed description.

Abbreviation	Detailed description
IFC	Inner fuel cycle
TEP	Tokamak exhaust processing system
ISS	Isotope separation system
SDS	Storage and delivery system
PCI	Pressure-composition isotherm
HMIE	Homomolecular isotope exchange reaction
ZPE	Zero-point energy
HID	Hydrogen-induced disproportionation reaction
DU	Depleted uranium
H	Protium
D	Deuterium
T	Tritium

Q	Hydrogen isotope
DSC	Differential scanning calorimetry
TPD	Temperature programmed desorption
TDS	Thermal desorption spectroscopy
α	Separation factor
Q_l	Atomic fraction of the light isotope in a certain phase
Q_h	Atomic fraction of the heavy isotope in a certain phase
T_{cr}	Critical temperature for zero isotope effect
$T_{1\text{ bar}}$	Operating temperature for atmospheric pressure delivery
T_{gap}	Mismatch between T_{cr} and $T_{1\text{ bar}}$
E_a	Apparent activation energy for the desorption process
T_p	Peak temperature of DSC curve
β	Heating rate
A	Pre-exponential factor
P_{eq}	Equilibrium plateau pressure
ΔH	Enthalpy change
ΔS	Entropy change
Z	Partition function for hydrogen isotopes in a certain phase
ω_H	Vibration frequency of interstitial hydrogen
\hbar	Reduced Planck constant

Reviewer #2

The manuscript describes the importance of thermodynamic and kinetic isotope effect on Ti-Pd substituted ZrCo alloy system in general and the $Zr_{0.8}Ti_{0.2}Co_{0.8}Pd_{0.2}$ alloy in particular. Various types of experimental methods are employed to examine both the effects. Further, the screening of Ti and Pd alloy substitution was carried out on the basis of computational method. The results are promising to reduce the temperature gap (T_{gap}) between the critical temperature (T_{cr}) and the operational temperature of hydrogen isotope delivery ($T_{1\text{ bar}}$).

Reply: Thanks for the reviewer's positive comment for research methods and results. Ti-Pd co-doping in the $Zr_{0.8}Ti_{0.2}Co_{0.8}Pd_{0.2}$ alloy enables a narrowed T_{gap} of 83.99 °C and a reduced kinetic barrier difference of 2.93 kJ/mol (correspondingly, 251.62 °C and 14.13 kJ/mol for ZrCo), and more importantly, a reduction of 30.18% in the maximum of isotope composition fluctuation. This proves the novelty and advancement of our proposed coordination design strategy that couples thermodynamic destabilization with vibration enhancement of interstitial isotopes for rational isotope engineering.

However, these studies are not new in entirety. The concepts of development of hydrogen isotope storage and delivery system based on ZrCo alloy systems are well documented. Hence, this manuscript does not provide any new insight into the science and technology directional development in this field. Hence, it may not be suitable for publication in Nature Communications.

Reply: Thanks for the reviewer's comment.

We agree with the reviewer's point that the concepts of development of hydrogen isotope storage and delivery system based on ZrCo alloy systems are well documented. Previously, the effects of element doping, nanostructuring, surface treatment and mechanical ball-milling on the anti-disproportionation and cycling stability for ZrCo alloy have been reported and studied extensively [*Advanced materials* 35 (2023); *Journal of Alloys and Compounds* 932 (2023)]. The increased kinetic energy barrier difference between dehydrogenation and hydrogen-induced disproportionation reaction (HID),

reduction of H(8e) occupation, extended bond length between H(8e) and neighbor Zr, mitigated lattice distortion and structural defects account for the durability improvement. Shui's group revealed a defect-derived disproportionation mechanism and reported a nano-single-crystal strategy. Owing to the low defect density, single-crystal ZrCo nanoparticles show significantly improved anti-disproportionation performances and cyclability [*Nature Communications* 14 (2023)]. **Recent advances in disproportionation mechanism have greatly improved anti-disproportionation ability and cycling stability, yet concurrently achieving subtle hydrogen isotope effect in the ZrCo-Q systems (Q=H, D, T) remains challenging.** Specifically, prompt availability to fuel streams with stable and controllable isotope composition is the key to supporting the tritium self-sufficiency and operational safety of fusion reactors. Nevertheless, the precise control of fuel D/T ratio is always disturbed by the prominent hydrogen isotope effect that originates from the isotopic affinity difference between gas and solid phases.

The thermodynamic isotope effect of ZrCo alloy exhibits obvious temperature dependence, and its inversion with temperature is confirmed by previous researches. Nevertheless, the specific critical temperature remains elusive due to the scattered thermodynamic parameters, which leads to a controversial understanding of thermodynamic isotope effect. From the kinetic point of view, the noticeable fluctuation of isotope composition during the desorption process but negligible during the absorption process with various H/D ratio were observed for ZrCo-based alloys at practical scales, which hinders the stable supply with desirable isotopic ratio. Owing to the prominent kinetic isotope effect, the isotopic composition deviation could be up to 7.30% during temperature programmed desorption with initial H/D=1/1 in a full-scale ZrCo bed assembly.

Existing studies were focused primarily on the doping effect of limited elements on the hydrogen isotope effect. However, the relationship between thermodynamic and kinetic isotope effects has been largely ignored in much of the previous literature. Moreover, the availability and enhancement of optimized alloys for isotope effect are not convincing without taking practical engineering condition into account (e.g., mixed isotopes, delivery

temperatures and times, etc.). Due to deficient understanding of the underlying interaction between the interstitial hydrogen isotope atoms and alloy matrix, the trial-and-error strategy still dominates the alloy development.

The novelty, originality and advancement of our work can be highlighted from the following aspects:

(1) Through elucidating the hydrogen isotope effect of ZrCo-based alloys for both absorption and desorption processes with comprehensive measurements over a wide temperature range, **we firstly proposed a comprehensive evaluation indicator (T_{gap})**, integrating the isotope effect with the thermodynamic stability of interstitial isotope atoms, which bridges the gap between practical engineering condition and intrinsic material properties.

(2) Based on the Van't Hoff relationship and harmonic oscillator model, **we demonstrated a coordination design strategy that couples thermodynamic destabilization with vibration enhancement of interstitial isotopes for rational isotope engineering** which guides the exploration of isotope storage and delivery material for fusion fuel cycle applications and, more in general, for applications related to the hydrogen technologies thought for supporting the energy transition.

(3) Inspired by the implications of the strategy, Ti and Pd elements were screened out and **a novel Ti-Pd co-doped ZrCo-based alloy ($\text{Zr}_{0.8}\text{Ti}_{0.2}\text{Co}_{0.8}\text{Pd}_{0.2}$) was designed to enable impressive success with a trivial isotope effect**, which outperforms previously reported ZrCo-based alloys (Fig. R1) and achieves a breakthrough in suppressing hydrogen isotope effect under practical engineering condition.

Fig. R1 Comparison of T_{gap} for ZrCo-based alloys in this work with literature data which are calculated from the thermodynamic parameters previously reported.

References:

- [R1] Jat, R. A., Parida, S. C., Agarwal, R. & Kulkarni, S. G. Effect of Ni content on the hydrogen storage behavior of ZrCo_{1-x}Ni_x alloys. *Int. J. Hydrogen Energy* **38**, 1490-1500, doi:<https://doi.org/10.1016/j.ijhydene.2012.11.053> (2013).
- [R2] Jat, R. A., Parida, S. C., Agarwal, R. & Ramakumar, K. L. Investigation of hydrogen isotope effect on storage properties of Zr–Co–Ni alloys. *Int. J. Hydrogen Energy* **39**, 14868-14873, doi:<https://doi.org/10.1016/j.ijhydene.2014.07.045> (2014).
- [R3] Jat, R. A. *et al.* Structural and hydrogen isotope storage properties of Zr–Co–Fe alloy. *Int. J. Hydrogen Energy* **40**, 5135-5143, doi:<https://doi.org/10.1016/j.ijhydene.2015.02.094> (2015).
- [R4] Monea, B. F. *et al.* Synthesis, characterization, and hydrogen isotope storage properties of Zr_{1-x}Ti_xCo and Zr_{1-x}Hf_xCo alloys (x = 0.1, 0.2). *Fusion Sci. Technol.* **77**, 382-390, doi:<http://doi.org/10.1080/15361055.2021.1903782> (2021).
- [R5] Jat, R. A., Pati, S., Parida, S. C., Agarwal, R. & Mukerjee, S. K. Synthesis, characterization and hydrogen isotope storage properties of Zr–Ti–Co ternary alloys. *Int. J. Hydrogen Energy* **42**, 2248-2256, doi:<https://doi.org/10.1016/j.ijhydene.2016.10.079> (2017).
- [R6] Jat, R., Rawat, D. & Sharma, A. Isotope effect on hydrogen storage properties of Ti and Nb co-substituted ZrCo alloy. *Int. J. Hydrogen Energy* **48**, doi:<http://doi.org/10.1016/j.ijhydene.2023.01.161> (2023).
- [R7] Jat, R. A., Rawat, D., Sharma, A. & Parida, S. C. Remarkable enhancement in durability of ZrCo alloy against hydrogen induced disproportionation by Ti and Nb co-substitution. *Int. J. Hydrogen Energy* **48**, 7431-7441, doi:<https://doi.org/10.1016/j.ijhydene.2022.11.086> (2023).

In a word, as the Reviewers' positive comments, the extensive and detailed experiment results are consistent with our proposed design strategy which gives a solid base to develop advanced isotope storage and delivery materials with subtle isotope effect. Since the established indicator and coordination design strategy combine the practical engineering condition and atomic interaction mechanism, it is of great significance from the scientific and technological point of view.

To strengthen the scope of our work, we presented additional work followed by the suggestions of the Reviewer #1 and #3 especially in terms of cycling stability, accurate

thermodynamic parameter acquisition, clarity in the presentation of the HD calibration, interpretation for negligible kinetic isotope effect during the absorption process.

In the revised manuscript, we added:

Page 6: The noticeable fluctuation of isotope composition during the desorption process but negligible during the absorption process with various H/D ratio were observed for ZrCo-based alloys at practical scales, which hinders the stable supply with desirable isotopic ratio. Owing to the prominent kinetic isotope effect, the isotopic composition deviation could be up to 7.30% during temperature programmed desorption with initial H/D=1/1 in a full-scale ZrCo bed assembly³³.

Page 6: Bridging the gap between the operating temperature ($T_{1 \text{ bar}}$) for atmospheric pressure delivery and the critical temperature (T_{cr}) for zero isotope effect is expected to make impressive gains in delivery performances. It is highly desirable to reduce the $T_{1 \text{ bar}}$ for narrowing the temperature gap and concurrently improving anti-disproportionation ability and cycling stability. On the other hand, the remarkable atomic mass distinction among hydrogen isotope atoms results in the zero-point energy (ZPE) difference in both gas and solid phases, dictating the interphase isotopic distribution and further the T_{cr} ^{34,35}.

Reviewer #3

The prominent isotope effect of ZrCo alloy originating from the most significant mass difference among hydrogen isotopes induces the isotope composition deviation in the fusion fuel gas. The manuscript presents comprehensive modeling and extensive characterization analysis to demonstrate the intrinsic reasons for the difference in the isotope effects between hydrogen isotope absorption and desorption processes. Authors recognize that the kinetic hydrogen isotope effect during the desorption process dominates the isotope composition fluctuation for ZrCo alloy. The manuscript presents a local coordination design strategy for minimizing the hydrogen isotope effect of ZrCo-based alloys during the desorption process. Therefore, they propose the novel Ti-Pd co-doped $Zr_{0.8}Ti_{0.2}Co_{0.8}Pd_{0.2}$ alloy that achieves more consistent isotopic behavior through thermodynamic destabilization with vibration enhancement of interstitial isotopes. As a result, it demonstrates that the alloys developed following this principle show minor isotope composition deviation during the desorption process. The manuscript is clearly written, well organized. As such, I believe this manuscript stands out from many other manuscripts in this area and can be recommended for publication after clarifying the questions and concerns raised below:

Reply: Thanks for the reviewer's positive comment on our work.

1. The authors point to the importance of accurate thermodynamic parameters with regard to the determination of thermodynamic isotope effect, especially for the separation factor (α) and further critical temperature (T_{cr}) for zero isotope effect. However, the equilibrium plateau pressures of PCI curves get slopped with the increase of the amount and type of substitution elements (Fig. S13-S16). How the authors define the equilibrium plateau pressure with a strong slopped curve?

Reply: Thanks for the reviewer's great comment.

It is necessary to obtain accurate de-/hydrogenation thermodynamic parameters for hydrogen isotope storage and delivery. In this work, the Van't Hoff equation based on the equilibrium pressures at different temperatures is adopted to calculate the thermodynamic

parameters for the isotope absorption and desorption processes. Due to various local atomic configurations produced by element doping, plateau region gets sloped with the amount of doping elements. As shown in Fig. R2, we can obtain the intersection points of tangent lines of $\alpha+\beta$ region (l_2) and α region (l_1) and tangent lines of $\alpha+\beta$ region (l_2) and β region (l_3) from the PCI curve, respectively. The equilibrium pressure can be further determined by the midpoint (point B) between two intersection points (point A and C). For a PCI curve with dual plateau (Fig. R3), the intersection points of tangent lines of low plateau $\alpha_1+\beta_1$ region (l_2) and α_1 region (l_1), tangent lines of low plateau $\alpha_1+\beta_1$ region (l_2) and intermediate hydride β_1 region (l_3), tangent lines of high plateau $\beta_1+\beta_2$ region (l_4) and intermediate hydride β_1 region (l_3), and tangent lines of high plateau $\beta_1+\beta_2$ region (l_4) and β_2 region (l_5) can be obtained, respectively. The equilibrium pressures can be further determined by the midpoints (point B_1 and B_2) between the intersection points (point A_1 and C_1 , point A_2 and C_2).

Fig. R2 Schematic of determining equilibrium pressures for sloping PCI curves with a single plateau.

Fig. R3 Schematic of determining equilibrium pressures for sloping PCI curves with a dual plateau.

2. L126-129, “Such kinetic isotope effect for the absorption process benefits rapid isotope storage, resulting from two aspects: (i) the significant driving force originating from the system pressure ($\sim 10^5$ Pa) far beyond the equilibrium plateau pressure (10^{-3} Pa)”, the negligible kinetic isotope effect of the designed ZrCo-based alloys during the absorption process should be further demonstrated by calculating the equilibrium plateau pressure at room temperature (20 °C).

Reply: Thanks for the reviewer’s constructive comment.

We have calculated the equilibrium plateau pressure of the $Zr_{0.8}Ti_{0.2}Co_{0.8}Pd_{0.2}$ alloy at room temperature (20 °C) to further demonstrate the negligible kinetic isotope effect due to the significant driving force caused by the substantial gap between system pressure and equilibrium pressure.

In the revised manuscript, we added:

Page 21: Based on the PCIs curves (Fig. S14-17), the corresponding thermodynamic parameters of the isotope absorption and desorption processes for the $Zr_{0.8}Ti_{0.2}Co_{0.8}Pd_{0.2}$ alloy were listed in the Table S5 and Table 3. The salient driving force caused by the substantial gap between system pressure and equilibrium plateau pressure at room temperature (0.136 Pa at 20 °C) that fitted from the Van't Hoff equation ensures the rapid storage kinetics and thus tiny kinetic isotope effect. For the desorption process, the minor difference of the $T_{1\text{ bar}}$ between the $Zr_{0.8}Ti_{0.2}Co_{0.8}Pd_{0.2}$ and $Zr_{0.8}Ti_{0.2}Co$ alloys suggested the Pd substitution for Co had little influence on the stability of the hydride phase.

In the revised supporting information, we added:

Table S5 Thermodynamic parameters for the protium/deuterium absorption processes of $Zr_{0.8}Ti_{0.2}Co_{0.8}Pd_{0.2}$ alloy.

Parameters	Absorption			
	Low plateau		High plateau	
	Q=H	Q=D	Q=H	Q=D
ΔH (kJ·mol ⁻¹ Q ₂)	-66.87	-67.44	-47.00	-54.90
ΔS (J·K ⁻¹ ·mol ⁻¹ Q ₂)	-115.90	-118.97	-105.91	-117.72

3. Explain how to quantitatively determine the isotope ratio during the whole desorption process according to the TDS curves, especially for the HD calibration. And please provide a table including the detailed data of Fig. 5f.

Reply: Thanks for the reviewer's great comment.

The desorption process of hydride phase with mixed isotope is stated in the manuscript "Initially, the isotope atoms randomly located in the lattice interstice of hydride phase. With temperature increasing, isotope atoms were excited to diffuse from bulk to subsurface and further surface. Combination of isotope atoms occurred on the surface and the molecules of different isotopic modifications were then released into the gas phase. The subsequent isotope exchange process was involved with the interphase isotope exchange between gas and solid phases and homomolecular isotope exchange reaction (HMIE) in the gas." The TDS profiles indicate that three hydrogen isotopologues including H₂, HD and D₂ are all detected during release process. The H₂ and D₂ calibration is made by a series of leakage rates using standard leak with high-purity H₂ and D₂ stream (>99.999%). The HD calibration can be achieved based on the law of hydrogen isotope mass conservation and negligible hydrogen isotope effect of ZrCo-based alloys during the absorption process. Specifically, the amount of hydrogen isotope in the hydride phase can be calculated by combining the ideal gas law and pressure drop under isochoric process using the Sievert's apparatus. The trivial kinetic hydrogen isotope effect during the absorption process allows us to determine the ratio of hydrogen isotope in the hydride phase and further deduce the amount of HD in the gas by H₂ and D₂ calibration. The detailed hydrogen concentration at certain temperatures are list in the Table S4.

In the revised manuscript, we added:

Page 20: To quantitatively track the isotope composition during the whole desorption process, the intensity of H₂, HD and D₂ species were calibrated. Especially, the HD calibration could be achieved based on the law of hydrogen isotope mass conservation and negligible hydrogen isotope effect of ZrCo-based alloys during the absorption process. The temperature dependence of the hydrogen concentration of accumulative release gas

during the desorption process was determined.

Page 20: The detailed data at certain temperatures are list in the Table S4.

In the revised supporting information, we added:

Table S4 Cumulative hydrogen concentration of release gas at certain temperatures during the whole desorption process.

Temperature (°C)	Hydrogen concentration (%)		
	ZrCo	Zr _{0.8} Ti _{0.2} Co	Zr _{0.8} Ti _{0.2} Co _{0.8} Pd _{0.2}
150	54.44	53.55	53.10
200	53.78	53.00	52.28
250	53.34	52.27	50.21
300	51.92	51.51	49.26
350	50.02	49.85	49.24
400	49.93	49.52	49.40
450	49.98	49.67	49.60
500	50.05	49.82	49.60
550	50.07	49.85	49.62
600	50.05	50.02	49.82
650	50.04	50.01	50.02

4. Measure the thermodynamic parameters of ZrCo_{0.8}Pd_{0.2} alloy to understand the explicit contribution of single Pd element doping.

Reply: Thanks for the reviewer's constructive comment.

We have prepared the ZrCo_{0.8}Pd_{0.2} alloy by induction levitation melting method. To understand the explicit contribution of single Pd element doping, the thermodynamic parameters of the ZrCo_{0.8}Pd_{0.2} alloy have been obtained based on a series of PCIs.

In the revised manuscript, we added:

Page 7: Based on the computational screening results, the Zr_{0.8}Ti_{0.2}Co_{0.8}Pd_{0.2} alloy was designed, prepared, and compared with the Zr_{0.8}Ti_{0.2}Co and ZrCo_{0.8}Pd_{0.2} alloys. Ti-Pd co-doping enables further suppressed isotope effect than Ti and Pd single doping. Specially,

the T_{gap} of the $\text{Zr}_{0.8}\text{Ti}_{0.2}\text{Co}_{0.8}\text{Pd}_{0.2}$ alloy reaches 83.99 °C while that of the ZrCo, $\text{Zr}_{0.8}\text{Ti}_{0.2}\text{Co}$ and $\text{ZrCo}_{0.8}\text{Pd}_{0.2}$ alloys are 251.62, 172.00 and 197.41 °C, respectively.

Page 22: To understand the explicit contribution of Pd single doping for hydrogen isotope effect, the $\text{ZrCo}_{0.8}\text{Pd}_{0.2}$ alloy was prepared. The single B2 phase, hierarchical layered microstructure at the micro/nanometer scales and preferred orientation of (110) plane were also demonstrated by XRD, SEM and TEM (Fig. S18 and Fig. S19a, b). Additionally, the STEM-HAADF image and corresponding EDS mapping indicate the homogeneous distribution and accurate content of alloy elements (Fig. S19c, d). The thermodynamic parameters of the $\text{ZrCo}_{0.8}\text{Pd}_{0.2}$ alloy were obtained based on a series of PCIs (Fig. S20-23) and listed in the Table S6. Single plateau and nearly consistent T_{cr} for both absorption and desorption processes suggest the reversibility of phase transition and verify the reliability to deduce the isotopic behavior from the reversible reaction. The T_{gap} for the $\text{ZrCo}_{0.8}\text{Pd}_{0.2}$ alloy is 197.41 °C, of which the reduction is mainly contributed to the increase of T_{cr} from 157.11 °C for the ZrCo alloy to the 193.31 °C for the $\text{ZrCo}_{0.8}\text{Pd}_{0.2}$ alloy.

Page 23: More importantly, Ti-Pd co-doping synergistically induces vibration enhancement, *i.e.*, the increased T_{cr} as well as thermodynamic destabilization, *i.e.*, the decreased $T_{1\text{ bar}}$, within the thermodynamically stable zone (Zone II), which remarkably narrowed the T_{gap} from 251.62 °C (ZrCo alloy) to 83.99 °C ($\text{Zr}_{0.8}\text{Ti}_{0.2}\text{Co}_{0.8}\text{Pd}_{0.2}$ alloy).

Page 26: Although decreased T_{gap} s were attained as expected for the $\text{Zr}_{0.8}\text{Ti}_{0.2}\text{Co}$ and $\text{ZrCo}_{0.8}\text{Pd}_{0.2}$ alloys, the reductions were primarily attributed to the decrease of $T_{1\text{ bar}}$ and increase of T_{cr} , respectively. To exploit the potential synergistic effect of Ti and Pd for suppressing hydrogen isotope effect, a rationally designed $\text{Zr}_{0.8}\text{Ti}_{0.2}\text{Co}_{0.8}\text{Pd}_{0.2}$ alloy was fabricated.

In the revised supporting information, we added:

Fig. S18 XRD pattern and Rietveld refinement result of $ZrCo_{0.8}Pd_{0.2}$ alloy.

Fig. S19 (a) SEM image, (b) TEM and corresponding HRTEM images, (c) STEM-HAADF image with EDS, (d) Quantitative content of alloy elements of $ZrCo_{0.8}Pd_{0.2}$ particles.

Fig. S20 PCI curves of $ZrCo_{0.8}Pd_{0.2}$ alloy for the protium absorption process.

Fig. S21 PCI curves of $\text{ZrCo}_{0.8}\text{Pd}_{0.2}$ alloy for the deuterium absorption process.

Fig. S22 PCI curves of $\text{ZrCo}_{0.8}\text{Pd}_{0.2}$ alloy for the protium desorption process.

Fig. S23 PCI curves of $\text{ZrCo}_{0.8}\text{Pd}_{0.2}$ alloy for the deuterium desorption process.

Table S6 Thermodynamic parameters for the protium/deuterium absorption and desorption processes of ZrCo_{0.8}Pd_{0.2} alloy.

Parameters	Absorption		Desorption	
	Q=H	Q=D	Q=H	Q=D
ΔH (kJ·mol ⁻¹ Q ₂)	-84.14	-88.74	85.72	89.25
ΔS (J·K ⁻¹ ·mol ⁻¹ Q ₂)	-133.83	-143.77	129.15	136.72
$T_{1 \text{ bar}}$ (° C)	/	/	390.72	379.79
T_{cr} (° C)		189.78		193.31
T_{gap} (° C)		/		251.62

REVIEWERS' COMMENTS

Reviewer #1 (Remarks to the Author):

The manuscript has been revised in agreement with the comments and the suggestions raised. It can be published.

Reviewer #2 (Remarks to the Author):

The authors have now improved the manuscript with clearly bringing out the novelty of the work. Most of the reviewers suggestions are incorporated in the revised manuscript. The manuscript can be accepted.

Reviewer #3 (Remarks to the Author):

The authors have revised this paper according to the reviewer's comments. I'd like to recommend its acceptance without alternation.